# Osmoregulatory Plasticity of Juvenile Greater Amberjack (*Seriola dumerili*) to Environmental Salinity

**DOI:** 10.3390/ani11092607

**Published:** 2021-09-05

**Authors:** Andre Barany, Neda Gilannejad, María Alameda-López, Liliana Rodríguez-Velásquez, Antonio Astola, Gonzalo Martínez-Rodríguez, Javier Roo, Jose Luis Muñoz, Juan Miguel Mancera

**Affiliations:** 1Department of Biology, Faculty of Marine and Environmental Sciences, Instituto Universitario de Investigación Marina (INMAR), Campus de Excelencia Internacional del Mar (CEI·MAR), University of Cádiz, Puerto Real, E11510 Cádiz, Spain; malamedalpezpez@gmail.com (M.A.-L.); lrodriguezv@unal.edu.co (L.R.-V.); juanmiguel.mancera@uca.es (J.M.M.); 2Instituto de Ciencias Marinas de Andalucía, Consejo Superior de Investigaciones Científicas (ICMAN-CSIC), Puerto Real, E11519 Cádiz, Spain; negi@norceresearch.no (N.G.); gonzalo.martinez@csic.es (G.M.-R.); 3NORCE Norwegian Research Centre AS, Uni Research Environment, Nygårdsgaten 112, E5008 Bergen, Norway; 4Department of Biomedicine, Biotechnology, and Public Health, Faculty of Sciences, Campus de Excelencia Internacional del Mar (CEI-MAR), University of Cadiz, Puerto Real, E11510 Cádiz, Spain; antonio.astola@uca.es; 5Grupo de Investigación en Acuicultura (GIA), Instituto Universitario Ecoaqua, Universidad de Las Palmas de Gran Canaria, Crta. Taliarte s/n, E35214 Gran Canaria, Spain; javier.roo@ulpgc.es; 6Department of Production, IFAPA Centro “El Toruño”, Junta de Andalucía, El Puerto de Santa María, E11500 Cádiz, Spain; jluis.munoz@juntadeandalucia.es

**Keywords:** intestine, ion transport, metabolism, Na^+^/K^+^-ATPase, osmoregulation, Ussing chambers

## Abstract

**Simple Summary:**

The greater amberjack, *Seriola dumerili*, is of particular interest for pisciculture diversification due to its flesh quality and worldwide market acceptance. Moreover, this species shows rapid growth at the juvenile stage and 75% survival under captivity conditions. Among growing conditions, salinity is a critical factor for optimal growth. This study specifically assessed and characterized the physiological status of osmoregulation in individuals acclimated to three experimental salinities (15, 22, and 36 psu) in land-based recirculating aquaculture systems for 92 days. The results indicated several physiological adaptations to the different environmental salinities, both at the osmoregulatory and metabolic levels. Overall, our results suggest a beneficial effect of lower salinities for the juvenile stage to improve growth performance and avoid seawater-parasite outbreaks prior to transfer to sea cages.

**Abstract:**

Osmotic costs in teleosts are highly variable, reaching up to 50% of energy expenditure in some. In several species, environmental salinities close to the isosmotic point (~15 psu) minimize energy demand for osmoregulation while enhancing growth. The present study aimed to characterize the physiological status related to osmoregulation in early juveniles of the greater amberjack, *Seriola dumerili,* acclimated to three salinities (15, 22, and 36 psu). Our results indicate that plasma metabolic substrates were enhanced at the lower salinities, whereas hepatic carbohydrate and energetic lipid substrates decreased. Moreover, osmoregulatory parameters, such as osmolality, muscle water content, gill and intestine Na^+^-K^+^-ATPase activities, suggested a great osmoregulatory capacity in this species. Remarkably, electrophysiological parameters, such as short-circuit current (Isc) and transepithelial electric resistance (TER), were enhanced significantly at the posterior intestine. Concomitantly, Isc and TER anterior-to-posterior intestine differences were intensified with increasing environmental salinity. Furthermore, the expression of several adeno-hypophyseal genes was assessed. Expression of *prl* showed an inverse linear relationship with increasing environmental salinity, while *gh* mRNA enhanced significantly in the 22 psu-acclimated groups. Overall, these results could explain the better growth observed in *S. dumerili* juveniles kept at salinities close to isosmotic rather than in seawater.

## 1. Introduction

The greater amberjack, *Seriola dumerili*, is a pelagic and epibenthic carnivore species that belongs to the Carangidae family and inhabits near- and off-shore waters with a vast worldwide distribution, comprising areas of the Atlantic and Indo-Pacific Oceans, as well as the Mediterranean Sea [1]. This species is highly appreciated commercially worldwide. It is also an important recreational pelagic fishery resource in Europe and North America. The total worldwide catch of the greater amberjack has increased more than ten-fold since 1990, from which 80% has been from the Mediterranean and Black Seas [1].

Overall, species of the genus *Seriola* (including *S. dumerili*) show rapid growth during the first years of life, either under wild, ~0.8 kg year^−1^ [1], or captivity conditions, ~2 kg year^−1^ [2]. This species has aroused strong interest in the aquaculture industry because of its fast growth as well as the high quality and price of its flesh [1,3,4]. Although the aquaculture of *Seriola* species has sharply increased in recent decades, its production is constrained by the reduced number of hatcheries and grow-out installations [3]. Japan is the largest aquaculture producer, and most of the amberjack production is based on imported and wild capture of juveniles for fattening.

Among culture conditions, temperature, salinity, stocking density, and feeding are critical factors for achieving optimal growth [4,5,6,7]. In teleosts, osmoregulatory processes have a high energy cost, varying from 20% to 50% depending on the species [8]. In this regard, the greater amberjack is grown to the commercial size (40 cm length/3–5 kg body mass) in about 24–36 months, at salinities between 14 and 38 psu [1,2,9]. It has been shown that salinities below 26 psu (to 15 psu) significantly improved growth performance in *S. lalandi* juveniles reared in RAS [9]. Our Research Group described similar results in a previous experiment where the growth of *S. dumerili* was tested with respect to different environmental salinities (15, 22, and 36 psu) [10].

Osmoregulatory maintenance is achieved by continuous replenishing of net water from imbibed seawater (SW, ~35 psu) desalination along the gastrointestinal tract [11,12]. Concomitantly to the desalination process, secretion of excess ions takes place across the gills [13], kidney [14], and as carbonate precipitates in the intestine [15,16]. The capability of regulating the volume of the cell in response to osmotic stress is known as homeostasis. To keep the allostatic maintenance [17], marine teleosts spend a significant amount of bioenergetic substrates (e.g., glucose, lactate, amino acids, etc.) in maintaining the plasma osmolality ~1/3 of the SW, and a specific ionic body fluid composition lower in divalent ions [15,16,18] *inter alia*. For this purpose, bioenergetics are mainly depleted by primary active transport, which requires ATP to move and exchange ions in order to energize passive transport across membranes of all osmoregulatory tissues [19,20]. Otherwise, these bioenergetic substrates could be relocated for growth, basal metabolism, and/or reproduction [8,21].

Several hormones are involved in the endocrine control of the teleost osmoregulatory processes at the adeno-hypophyseal level. Prolactin (PRL) is essential for freshwater adaption. Specifically, it has been shown to modify the physiological processes by reducing tissue permeability and active ion uptake [22]. It has also been established that PRL is an antagonist to the growth hormone (GH) regarding its effects on the ion secretory mechanisms [23]. Besides GH’s key role in controlling growth processes [24], it has also been reported as an SW-adapting hormone in synergy with cortisol and/or thyroid hormones [22,23,25]. Somatolactin (SL) is a pleiotropic hormone from the GH/PRL family, involved in a wide range of physiological processes (e.g., osmoregulation, metabolism, pigmentation, etc.), with species-specific roles [26,27]. The proopiomelanocortin (POMCs) precursors are produced in the adenohypophysis. POMCA is processed into alpha-melanocyte-stimulating hormone (α-MSH) in the melanotroph cells [28]. In contrast, the end product of POMCB is processed into adrenocorticotropic hormone (ACTH) in the corticotroph cells. Variable size POMC transcripts have been found in extra pituitary tissues, explained by alternative splicing [29]. Finally, ACTH stimulates the cortisol synthesis in the inter-renal cells of the head kidney, presenting a dual role in stress and osmoregulatory responses [28].

This study tested the influence of different environmental salinities (15, 22, and 36 psu) on *S. dumerili* osmoregulatory and metabolic status. Changes in plasma and hepatic metabolites were assessed. Modifications in plasma osmolality, as well as the Na+/K+-ATPase activity in the gills and the anterior and posterior regions of the intestine, were also evaluated. In addition, several electrophysiological parameters (transepithelial potential and short circuit current) were determined at different intestinal regions. Finally, the influence of different environmental salinities on mRNA expression levels for various adeno-hypophyseal hormones: *prolactin* (*prl*), *growth hormone* (*gh*), and *somatolactin* (*sl*), and hormonal precursors: *proopiomelacortin a* (*pomca*) and *proopiomelacortin b* (*pomcb*), involved in osmoregulatory processes, were examined.

## 2. Materials and Methods

### 2.1. Animals and Experimental Design

In a previous experiment, we assessed the influence of three different experimental salinities (15, 22, and 36 psu) on the growth of *S. dumerili* juveniles [11]. Briefly, greater amberjack (*S. dumerili*) juveniles, provided from a stock population at the University Research Institute of Sustainable Aquaculture and Marine Ecosystems, IU-ECOAQUA (Las Palmas de Gran Canaria, Canary Islands, Spain), were transferred to the IFAPA, Centro El Toruño experimental facilities (El Puerto de Santa María, Cádiz, Spain). After fish (n = 300; 95.1 ± 23.0 g body mass, 16.4 ± 1.5 cm fork length) acclimatization to the facilities’ conditions for 40 days, they were randomly distributed in groups of 25 into twelve 650 L-fiber glass tanks. Three different experimental salinities (15, 22, and 36 psu) were established in quadruplicate tanks in three independent 8 m^3^-RAS. The final stock density per replicate was 3 kg m^−3^, and all tanks were kept under continuous aeration, under 12 h of light/12 h of darkness, and ambient temperatures (17–23 °C) for 92 days (from March to May 2018). 

Experimental salinities were prepared by mixing the natural SW and dechlorinated municipal freshwater. For each RAS, a 5% daily water renovation was performed. Parameters such as salinity, temperature, pH, and oxygen concentration were monitored daily in each tank (data not shown). Fish were fed a total of 3% of their body mass with a commercial fish feed (Skretting, Burgos, Spain), distributed throughout the daylight, using continuous belt feeders. This study was performed in accordance with the Guidelines of the European Union (2010/63/UE) and the Spanish legislation (RD 1201/2005 and law 32/2007) for the use of laboratory animals.

### 2.2. Sampling Protocol

After the experimental period, six fish from each replicate (n = 12 per salinity) were randomly sampled to minimize tank effects. Fish were euthanized with a lethal dose of 0.7 mL of 2-phenoxyethanol (Sigma-Aldrich, Madrid, Spain, P1126) per L of the experimental water. Blood was collected from caudal vessels with ammonium-heparinized syringes (Sigma-Aldrich, H6279, 25,000 units/3 mL of saline 0.9% NaCl) and centrifuged (3 min, 13,030 *g*, 4 °C) to obtain plasma. The second gill arch on the dorsal side and intestinal tissue biopsies from the anterior (AI) and posterior (PI) regions were collected for determination of Na^+^/K^+^-ATPase activity (NKA) in Eppendorf tubes containing homogenization SEI buffer (150 mM sucrose, 10 mM EDTA, 50 mM imidazole, pH 7.3). For gill biopsies, filaments were excised for analysis, whereas the cartilaginous arch was removed. The intestinal tissue biopsies from two different sections were established as follows: (i) AI, corresponding to 1.5–2 cm after the end of the pyloric caeca; and (ii) PI, which corresponds to a section of distal intestine, 2–3 cm in length, delimited by the ileorectal valve that precedes the rectum. Further details on the morphology of the gastrointestinal tract of this species can be found in Grau et al. [30].

Liver was collected for metabolite analysis. Moreover, white muscle (~1 g) was collected from the posterior region to the anal vent for the determination of water content by desiccation. All samples were immediately frozen and stored at −80 °C until analysis. Pituitaries were placed in Eppendorf tubes containing 100 µL of RNAlater (Ambion^®^, Applied BioSystems). These samples were kept for 24 h at 4 °C and then stored at −20 °C until total RNA isolation was performed.

### 2.3. Plasma and Tissue Parameters

Osmolality was measured in 20 μL plasma samples using a Fiske One-Ten vapor pressure osmometer (Fiske Associates, Advanced Instruments, Norwood, MA, USA). To assess the metabolite levels, samples from the liver were individually minced on an ice-cold Petri dish and subsequently homogenized by mechanical disruption (Ultra-Turrax^®^, T25 basic with an S25N-8G dispersing tool, IKA^®^-Werke) with 7.5 vol. (*w*/*v*) of ice-cool 0.6 N perchloric acid, and neutralized after adding the same volume of 1 M KHCO_3_. After that, the homogenates were centrifuged (30 min, 3500 *g*, 4 °C), and the supernatants were recovered and separated in different aliquots. Note that for triglycerides, aliquots were made prior to centrifugation. The aliquots were then stored at −80 °C until used in metabolite assays. Metabolite concentrations in plasma (glucose, lactate, triglycerides, and proteins) and liver (glucose and triglycerides) were determined using commercial kits from Spinreact (Barcelona, Spain) (Glucose-HK Ref. 1001200; Lactate Ref. 1001330; Triglycerides ref. 1001311) with reactions adapted to 96-well microplates. Plasma total protein concentration was determined by a BCA Protein Assay Kit (PIERCE, Thermo Fisher Scientific, Waltham, MA, USA, #23225) using BSA as the standard.

Liver glycogen levels were assessed using the method from Keppler and Decker [31]. The glucose obtained by later glycogen breakdown (after subtracting free glucose levels) was determined using the commercial kit described above for glucose. Standards and samples were measured in duplicate. All the assays were run in duplicate in an automated microplate reader (PowerWave 340, BioTek Instrument Inc., Winooski, VT, USA) using KCjunior™ software. For muscle water content determination, ~1 g of caudal white muscle was blotted dry and weighed (wet mass) and then desiccated in an oven for 48 h at 60 °C until dehydrated to a stable weight (dry mass).

### 2.4. Na^+^/K^+^-ATPase Activity (NKA)

Gill and intestinal (from anterior and posterior regions) NKA activity were analyzed using an NADH-linked kinetic assay in a 96-well microplate run at 25 °C for 10 min, as described in McCormick, 1993 [32]. Frozen tissues were homogenized on ice SEID (0.1% sodium deoxycholate in SEI buffer, pH = 7.3) and centrifuged at 3200× *g* for 5 min at 4 °C. The supernatant was assayed for ATPase activity in the presence and absence of the NKA-specific inhibitor ouabain (0.5 mM, Sigma O3125). NADH oxidation was determined spectrophotometrically at 340 nm. The difference in absorbance between the inhibited and uninhibited assay mixtures was used to calculate the NKA-specific (ouabain-sensitive) activity, expressed as µmol ADP mg^−1^ protein h^−1^. Total protein was measured using the bicinchoninic acid protein assay (BCA) with bovine serum albumin (BSA) as standard (Thermo Scientific, Rockford, IL, USA). Both assays were run in duplicate in an automated microplate reader (PowerWave 340, BioTek Instrument Inc., Winooski, VT, USA) controlled by KCjunior™ software.

### 2.5. Ussing Chamber Experiments

Additional anterior and posterior intestines were collected fresh following the same regionalization established for NKA activity analysis, isolated, and mounted in Ussing chambers as previously described [16]. Briefly, the tissues were opened longitudinally, flattened, and placed between Ussing chambers containing 2 mL of serosal artificial *Seriola* spp. physiological saline (i.e., symmetrical conditions), each: 175 mM NaCl, 2 mM NaH_2_PO_4_, 10 mM NaHCO_3_, 3 mM KCl, 1.5 mM CaCl_2_, 1 mM MgSO_4_, 7 mM glucose; osmolality was adjusted to 390 mOsm kg^−1^ with mannitol and pH was adjusted to 7.55 with Trizma (T1503, Sigma-Aldrich). In vitro, the tissue was bilaterally gassed with humidified 0.5% CO_2_ in an oxygen mixture, and the temperature was maintained constant at 19 °C. Transepithelial potential (TEP, mV) was referenced to the luminal side, and short circuit current (Isc, μA cm^−2^) was monitored by clamping of epithelia to 0 mV and expressed as negative for either secretion of anions (serosal to luminal) or absorption of cations (luminal to serosal). The in vitro preparations were left undisturbed until reaching a steady-state (~30 min). Transepithelial electric resistance (TER, Ω·cm^2^) was manually calculated (Ohm’s law) using the current deflections induced by a ±1 mV pulse of 3 s every minute. Voltage clamping and current injections were performed using epithelial amplifiers DVC1000 (World Precision Instruments, Sarasota, FL, USA). All data were recorded onto a computer using a Lab-Trax-4 acquisition system (World Precision Instruments, Sarasota, FL, USA) and LabScribe3 (iWorx Systems Inc., Dover, NH, USA).

### 2.6. RNA Isolation and Quantitative Real-Time PCR

Whole pituitaries were individually processed for total RNA extraction using the NucleoSpin^®^ RNA XS kits (Macherey Nagel, Düren, Germany). A polytron PT 1200 E with a PT-DA 03/2EC-E050 (Kinematica AG, Malters, Switzerland) dispersing tool was used to homogenize them. Genomic DNA (gDNA) was removed via on-column DNase digestion at 37 °C for 30 min using rDNase (RNase-free) included within the kit. The RNA concentration was measured with a Qubit 2.0 fluorimeter and Qubit™ RNA BR assay kit (Invitrogene, Thermo Fisher Scientific, Walthem, MA, USA). RNA quality was tested in a Bioanalyzer 2100 with the RNA 6000 Nano kit (Agilent Technologies, Inc., Santa Clara, CA, USA). Only samples with an RNA integrity number greater than 8.0 were used for real-time quantitative PCR (qPCR). Total RNA (50 ng) from each sample was reverse-transcribed in a 20 μL reaction volume using the qScript™ cDNA synthesis kit (Quanta Bio, Beverly, MA, USA) in a Mastercycler^®^ proS (Eppendorf AG, Hamburg, Germany). Reactions were diluted 10-fold with 10 mM Tris-0.1 mM EDTA (pH = 8.0), to obtain final nominal concentrations of 250 pg μL^−1^. A pool of cDNAs from all the samples was used for calibration plots, using six 1/10 serial dilutions, from 1 ng to 10 fg, to assess the different primers linearity and efficiency combinations, as well as for inter-assay calibration. Control reactions with RNase-free water (NTC) and RNA (NRT) were included in the analysis to ensure the absence of primer-dimers and genomic DNA contaminations. For all primers pairs, linearities (R^2^) and amplification efficiencies were in the ranges of 0.999–1 and 90.6–100.1%, respectively. Previously, primers were tested for final working optimum concentrations (200 nM) and temperature (60 °C). Each reaction was performed in duplicate containing 0.5 μL of specific forward and reverse primers (Table 1), 5 μL PerfeCTa™ SYBR^®^ Green FastMix™ (Quanta Bio, Beverly, MA, USA), and 4 μL of cDNA (1 ng). Reactions were performed in a final volume of 10 μL using Hard-Shell^®^, white well/blue shell, Low-Profile, Thin-Wall, 96-Well, Skirted, PCR plates covered with Microseal^®^ B Adhesive Seals (BioRad, Hercules, CA, USA). The thermocycling procedures were carried out with an initial denaturation and polymerase activation at 95 °C for 10 min, followed by 40 cycles of denaturation for 15 s at 95 °C, annealing, and extension at 60 °C for 30 s, and finishing with a melting curve from 60 °C to 95 °C, increasing by 0.5 °C every five seconds. Melting curves were used to ensure that only a single PCR product was amplified and to verify the absence of primer-dimer artifacts. Relative gene expressions were quantified in a CFX Connect™ Real-Time PCR Detection System under the control and analysis of CFX Manager™ software (Bio-Rad), using the ΔΔC_T_ method [33], corrected for efficiencies [34], and normalized by geometric averaging of two references genes [35], actin beta (*actb*) and eukaryotic elongation factor 1 alpha (*eef1a*). These two reference genes were selected owing to their low variability (Mean CV = 0.0793, Mean M = 0.2287), complying with the requirements established by the GenNorm Target Stability Value (CV < 0.25 and M < 0.5). Primer sequences for qPCR, amplicon sizes, and GenBank accession numbers for respective genes are shown in Table 1.

In brief, primers for qPCR used in this study were designed from the cDNAs sequences obtained from a previous unpublished RNAseq analysis of *S. dumerili* in brain, pituitary, liver, and kidney samples obtained by our research group. For this analysis cDNA samples from the tissues mentioned above were sent to Bioarray (Elche, Alicante, Spain). The RNAseq was performed by NGS mass sequencing, using the Ion Total RNA-Seq Kit v2 in the Ion Proton Sequencer from Life Technologies. The sequences were assembled with the Trinity software, annotating the transcripts with the Blast2GO software. Primers were designed using Primer3 software v.0.4.0 (available at https://bioinfo.ut.ee/primer3-0.4.0/ (accessed on 10 June 2021) and were synthesized by Metabion International AG (Planegg, Bavaria, Germany).

### 2.7. Statistical Analysis

Statistical comparison for all given results was performed using either a one- or two-way ANOVA. All data are presented as the mean ±SEM (Standard Error of the Mean), except for fish stock’s body mass and fork length, represented as the mean ± SD (Standard Deviation). Prior to analyzes, both normality of distribution and homoscedasticity of variance were confirmed by Shapiro-Wilk and D’Agostino-Pearson tests, respectively. Besides, for all data sets, outliers were identified by the ROUT method at Q = 1%. A Tukey’s *post hoc* test followed all one-way ANOVA analyses when significant differences were detected, whereas a Bonferroni’s *post hoc* test followed the two-way ANOVA analysis. All statistical analyses were performed with GraphPad Prism 8.0 (GraphPad Software, La Jolla, CA, USA), and significance for all tests was set at *p* < 0.05.

## 3. Results

### 3.1. Plasma and Tissue Parameters

Plasma osmolality at 36 psu was significantly higher than at 15 and 22 psu (Figure 1A). Based on the plotted linear regressions of plasma osmolality against experimental salinities (R^2^ = 0.9976), the calculated isosmotic point in *Seriola dumerili* species was 10.4 psu for juveniles. Determination of the muscle water content showed no differences among experimental groups (Figure 1B).

Plasma glucose increased significantly at both extreme salinities, 15 and 36 psu (Table 2). Plasma lactate at 36 psu was significantly lower than at 15 psu, while at 22 psu, the values were intermediate (Table 2). Analysis of triglycerides showed no significant differences among experimental groups (Table 2).

In the liver, free glucose and glycogen at 36 psu were significantly higher than at 15 psu (Table 2). Hepatic triglycerides were significantly enhanced at the intermediate salinity (22 psu) with respect to other environmental salinities (15 and 36 psu) (Table 2).

### 3.2. Na^+^/K^+^-ATPase Activity (NKA)

Gill NKA activity presented a direct relationship enhancement with respect to environmental salinity, showing the highest value at 36 psu; i.e., 15 psu ≤ 22 psu < 36 psu (Figure 2). Similarly, NKA activity in the anterior intestine (AI) also showed this pattern of change, enhancing significantly at 36 psu with respect to the lowest salinity. In contrast, NKA activity in the posterior intestine (PI) was unaffected by environmental salinity (Figure 3). Remarkably, differences between AI and PI were significantly intensified with increasing environmental salinity. The significances of the two-way ANOVA for Figure 3 were as follows; P_salinity_ = 0.053; P_intestinal region_ < 0.0001; P_interaction_ = 0.0743.

### 3.3. Ussing Chamber Experiments

Transepithelial electrical resistance (TER, Ω cm^2^), under-voltage clamp to 0 mV, and symmetric conditions did not significantly change among experimental salinities. However, differences between AI and PI were intensified with increasing environmental salinity. Regardless of the environmental salinities, TER values were similar within intestinal regions at 138 and 249 Ω cm^2^ for AI and PI, respectively (Figure 4A). 

Although absorptive short-circuit current (Isc, μAmp cm^−2^) did not significantly differ among experimental salinities, differences between AI and PI were just significant at the highest environmental salinity (36 psu). In the AI, baseline Isc were −2.3 ± 1.4; −1.2 ± 1.4, and −0.7 ± 1.2 µA cm^−2^ at 15, 22 and 36 psu, respectively. In the PI baseline Isc were −4.1 ± 1.3; −3.1 ± 0.8, and −4.7 ± 1.0 µA cm^−2^ at 15, 22 and 36 psu, respectively (Figure 4B).

The positive lumen transepithelial potential and the negative Isc needed to clamp the TEP to zero is consistent with the intestinal epithelium being anion absorptive, regardless of the experimental salinity (Figure 4B). No significant differences were detected for TER nor Isc among different experimental salinities in a given intestinal region (Figure 4A,B). The results of the two-way ANOVA results for Figure 4 were as follows: (A) P_intestinal region_ < 0.0001; P_salinity_ = 0.0568; P_interaction_ = 0.6825; (B) P_intestinal region_ = 0.0122; P_salinity_ = 0.6802; P_interaction_ = 0.5850.

### 3.4. Gene Expression

Relative *prl* mRNA expression levels showed an inverse linear relationship with increasing environmental salinity, i.e., 15 psu > 22 psu > 36 psu) (Figure 5A). Moreover, gh expression enhanced significantly in the 22 psu-acclimated groups, whereas it decreased again at 36 psu when compared to 15 psu groups (Figure 5B). No significant differences were detected for *sl, pomca,* or *pomcb* mRNA expression levels (Figure 5C–E).

## 4. Discussion

Our study supports that the greater amberjack is a euryhaline species with high osmoregulatory capacities from full-strength SW (36 psu) down to isosmotic water (15 psu). Moreover, anterior-to-posterior regionalization is highlighted and is heavily influenced by environmental salinity. Specifically, we found significant basal differences that were intensified between AI and PI in NKA activity, TER, and Isc in relation to increasing environmental salinity.

Branchial and intestinal NKA enzymes are essential modulators of osmoregulatory processes and responsible for major bioenergetic expenditure during osmoregulation [8,36,37]. Marine teleosts (and lampreys) ingest large amounts of SW to compensate for the passive water loss, which is consecutively processed by the different intestinal regions until close to isosmotic levels. The driving force for NaCl absorption/secretion and overall ion movements across membranes is primarily achieved by the electrogenic basolateral NKA [38,39]. Our results show that NKA activity in gills and the AI is enhanced at higher environmental salinities in *S. dumerili*. These findings are in agreement with previous studies in other teleosts [37,40] and lampreys [41,42], where NKA activity was shown to increase in gills and the AI at higher salinities, contributing to the achievement of the osmoregulatory homeostatic state. Specifically, the anterior intestine is the primary entrance of monovalent ions, reducing the imbibed SW salinity up to 50%. In other species, this role might also be supported by the esophagus via passive mechanisms and not ATP-dependent enzymes, in contrast to the anterior region of the intestine [21,43,44]. Note that water permeability of the esophagus can also differ across species, but not for the anterior intestine, where moderate to high water absorption rates are kept across several studied species [45,46]. Independently of the followed pathway for monovalent ions removal from the proximal gastrointestinal tract, this excess in ions is later secreted throughout the gills [47].

To further characterize the effects of environmental salinity on the intestine, we used an electrophysiological approach to determine several bioelectrical parameters and their changes due to salinity. TER is an indicator of epithelial leakiness, thus reflecting tissue permeability to ions/molecules [48]. It has been previously shown that TER might change due to environmental salinity depending on the species. For example, intestinal TER was affected in gilthead seabream (*Sparus aurata*) and Atlantic salmon (*Salmo salar*) [43,44]. In contrast, in Senegalese flatfish sole (*Solea senegalensis*) and sea lamprey (*Petromyzon marinus*), it was not [40,49]. Although in *S. dumerili*, TER did not significantly vary among experimental salinities, the PI (or the middle intestine according to some authors) seems to have a greater selectivity (being less permeable) compared to the AI. Moreover, the most remarkable change in TER in this study was that differences between the AI and PI were intensified at 22 and 36 psu when compared to 15 psu. In this regard, the distal regions might act as final controllers of the osmoregulatory processes of the gastrointestinal tract [16]. Although such differences among regions (TER in the AI < PI) are also found in other marine species [15,50], the changes in the environmental salinity do not seem to be the main driving factor [16,40]. Additionally, based on the measurements of Isc, the net transcellular intestinal fluxes are purely absorptive for anions in both intestinal regions of this species. Remarkably is at the highest salinity (36 psu) when higher absorptive Isc were observed in the PI than in the AI, suggesting once more that the distal regions act as final controllers of the osmoregulatory processes in this tissue by showing functional differences between both regions [16,40].

Our results in *S. dumerili* juveniles acclimated to 36 psu showed a moderate increase in plasma osmolality (~17 mOsm kg^−1^ H_2_O), whereas muscle water content remained statistically unaffected. Remarkably, the muscle water content in acclimated fish at extreme salinities (15 and 36 psu) was more similar between them (~59%) than at intermediate salinity (~65%). These results suggest that this species is at intermediate salinities (between 16 and 34 psu), and according to better growth [13], where they perform a better water balance. Metabolic water is essential for physiological biochemistry that allows for homeostasis, metabolism, and growth, *inter alia*. In addition, its content might directly affect plasma osmolality, as shown for the highest salinity. Thus, for this species, the optimal range for balanced growth and osmoregulation is a salinity of around 22 psu. In *S. salar* acclimated to SW, a 3% decrease in muscle water content was observed, showing a final muscle water content of 76% [51]. Similarly, the muscle water content in SW-acclimated sea lamprey juveniles was also ~76% [41]. Tissue dehydration occurs in fish that are unable to osmoregulate in hyperosmotic environments. Thus, higher muscle hydration levels may be related to osmoregulation initial energetic demands, resulting in lower lipid and higher muscle water; a typical relationship is seen in teleosts [52].

Environmental salinity challenges impose energetic regulatory costs for active ion transport [8,37] that are compensated by bioenergetic metabolic expenditure [36,53]. Ultimately homeostasis is physiologically achieved principally by (i) continuous water replenishing from imbibed SW desalination along the gastrointestinal tract [12,38]; (ii) simultaneous secretion of ions excess across the gills [13] and kidney [14], and (iii) carbonate aggregates precipitation in the intestine [15,16]. In this regard, the present study revealed that *S. dumerili* juveniles acclimated to 36 psu depleted plasma availability of lactate and hepatic triglycerides preferentially. In contrast, hepatic free glucose and glycogen were significantly accumulated in the liver. Therefore, carbohydrates such as liver glycogen may be mobilized from the liver to the plasma at low salinities, whereas it is stored at higher salinities. Glucose is considered the best non-protein energy source, which is catabolized by glycolysis under aerobic conditions. Interestingly, in our study with *S. dumerili*, plasma hyperglycemia at 15 and 36 psu resulted in hepatic glycogen (glycogenesis) only at 36 psu, and at the same time correlated to higher salinities’ plasma lactate depletion [36,54]. Overall, osmotic acclimation induces differential preferences for specific bioenergetic metabolites regarding environmental salinity. These metabolic modifications, imposed by osmotic acclimation, could explain the better growth performance observed in *S. dumerili* juveniles acclimated to 15 and 22 psu compared to 36 psu [10].

To our knowledge, no studies have addressed the pituitary control of the osmoregulatory and growth processes in this species at the molecular level to date. In the present study, we found that *prl* expression, a pleiotropic hormone related to hyposmotic environment-adaption in fish, significantly decreased at the highest salinity, according to what has been observed in several other studies in teleosts [22]. In contrast, *gh* expression showed the highest levels at 22 psu, whereas its lowest was at 15 psu. Interestingly at 15 psu, *S. dumerili* showed slightly better growth performance [10]. In this manner, it has been shown that fish growth enhancement might be linked to decreased *gh* expression due to its negative feedback with increased IGF-I levels [55]. We also suggest that these differences could be due to a dual interaction of somatotropic growth and the osmoregulatory process. Several teleosts have previously shown that GH plays a role in osmotic acclimation and controls the growth and energy metabolism in fish [22]. In salmonids, GH acts as an SW-adapting hormone. However, in non-salmonid species, the evidence contradicts this fact by exhibiting an apparent hyposmoregulatory role in some species and no clear osmoregulatory role in others [22]. However, we cannot rule out that the final physiological effects might be the action of GH combining with other hormones such as cortisol [23]. Remarkably, although we did not find significant differences, we observed a trend to increase mRNA expression levels with increasing environmental salinity of the *sl*. Note that in *S. aurata*, a stimulatory effect of SL on lipogenic enzymes, which stimulate lipid mobilization, has been reported [56], suggesting a possible metabolic role of this hormone in *S. dumerili* associated with high salinity acclimation. Lastly, insignificant differences were observed in the expression of both proopiomelanocortin genes (*pomca* and *pomcb*), which would correlate with putative ACTH/cortisol and α-MSH increment. However, and due to the osmoregulatory role of cortisol in different teleost [26,27], further studies are necessary to establish the osmoregulatory role of ACTH/cortisol in *S. dumerili*.

## 5. Conclusions

Overall, our results confirm the hypothesis that, in the euryhaline *S. dumerili*, ambient salinity reorganizes the osmoregulatory tissues to compensate for the osmotic imbalance and strongly influences the physiological processes redistributing the available energy and affecting growth [8,9,10,52]. Moreover, our results suggested the influence of different environmental salinities on adeno-hypophyseal gene expression patterns encoding the growth and osmoregulatory performance. Together with previous studies in *Seriola* spp. related to optimization of the rearing conditions, these results provide new clues for in-land on growing greater amberjack in RAS, suggesting a beneficial effect of lower salinities for the juvenile stage to improve the growth performance [10] prior to their transfer to sea cages, but also potentially avoiding the SW parasite outbreaks (e.g., *Neobenedenia* spp.) by isosmotic baths in *Seriola* spp. [57,58].

## Figures and Tables

**Figure 1 animals-11-02607-f001:**
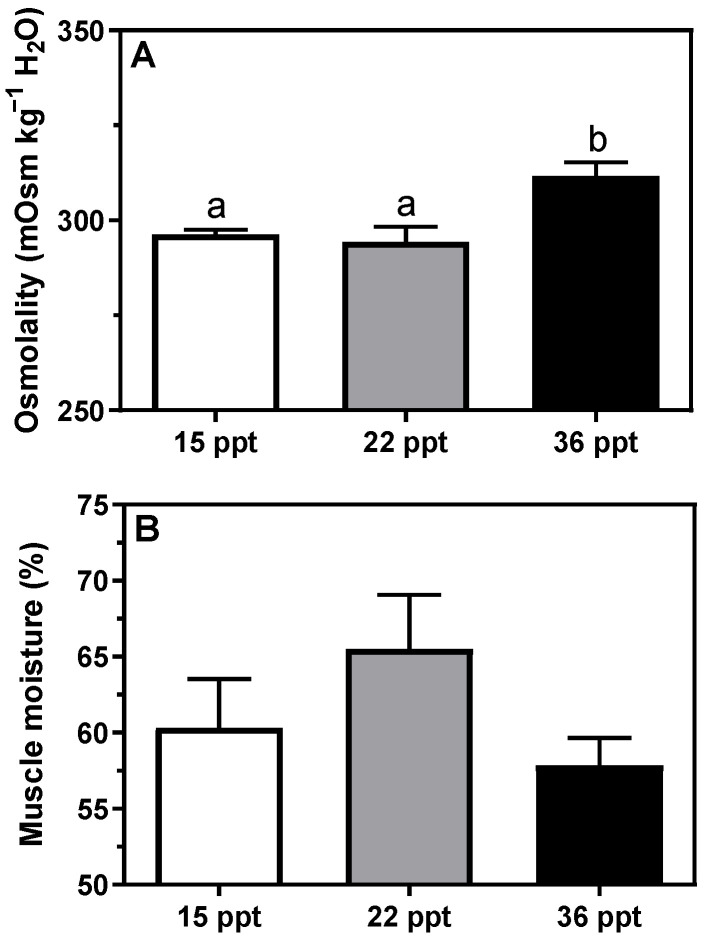
Changes in (**A**) plasma osmolality and (**B**) muscle water content in *S. dumerili* individuals acclimated to different environmental salinities (15, 22, and 36 psu). Data are presented as mean ±SEM (n = 10–11). Different superscript letters indicate significant differences among salinities (*p* < 0.05, one-way ANOVA followed by Tukey´s *post hoc* test).

**Figure 2 animals-11-02607-f002:**
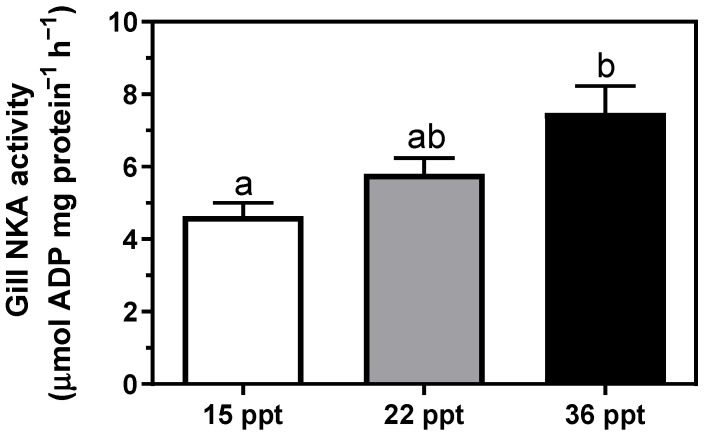
Gill NKA activity in *S. dumerili* individuals acclimated to different environmental salinities (15, 22, and 36 psu). Data are presented as mean ± SEM (n = 11–12). Different superscript letters indicate significant differences among salinities (*p* < 0.05, one-way ANOVA followed by Tukey´s *post hoc* test).

**Figure 3 animals-11-02607-f003:**
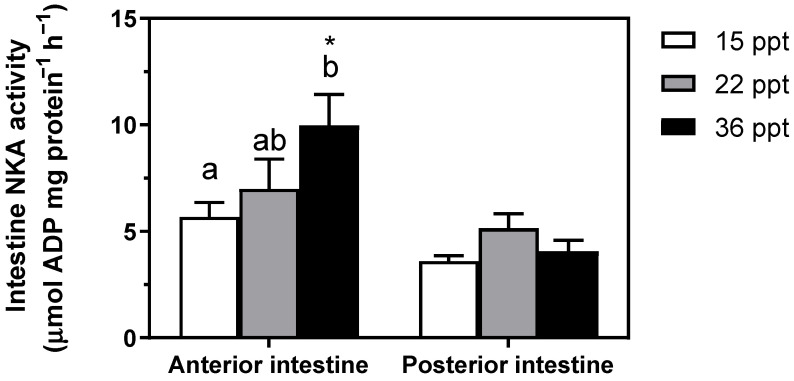
Intestine NKA activity in the anterior and posterior intestine in *S. dumerili* individuals acclimated to different environmental salinities (15, 22, and 36 psu). Data are presented as mean ±SEM (n = 7–12). Different superscript letters indicate significant differences among salinities in a given intestinal region. Asterisks (*) indicate significant differences between intestinal regions within a given salinity (*p* < 0.05, two-way ANOVA followed by Tukey´s *post hoc* test).

**Figure 4 animals-11-02607-f004:**
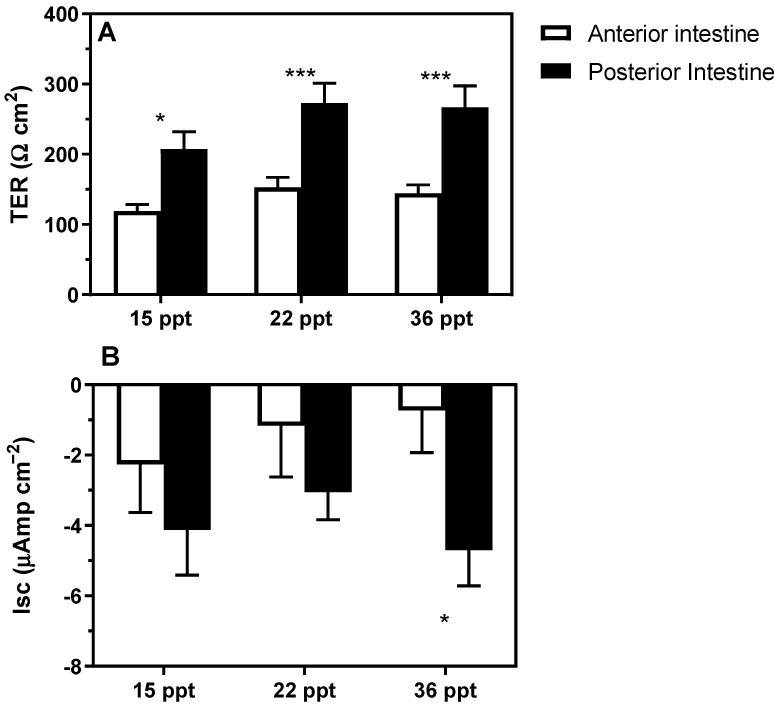
(**A**) Transepithelial electrical resistance (TER, Ω cm^2^) and (**B**) short-circuit current (Isc, μA cm^−2^) in the anterior and posterior intestine in *S. dumerili* individuals acclimated to different environmental salinities (15, 22, and 36 psu). Data are presented as mean ± SEM (n = 8). *Post hoc* significance levels are indicated by asterisks (*) used to indicate significant differences between intestinal regions within a given salinity (* *p* < 0.05, *** *p* < 0.001; two-way ANOVA followed by Bonferroni´s *post hoc* test).

**Figure 5 animals-11-02607-f005:**
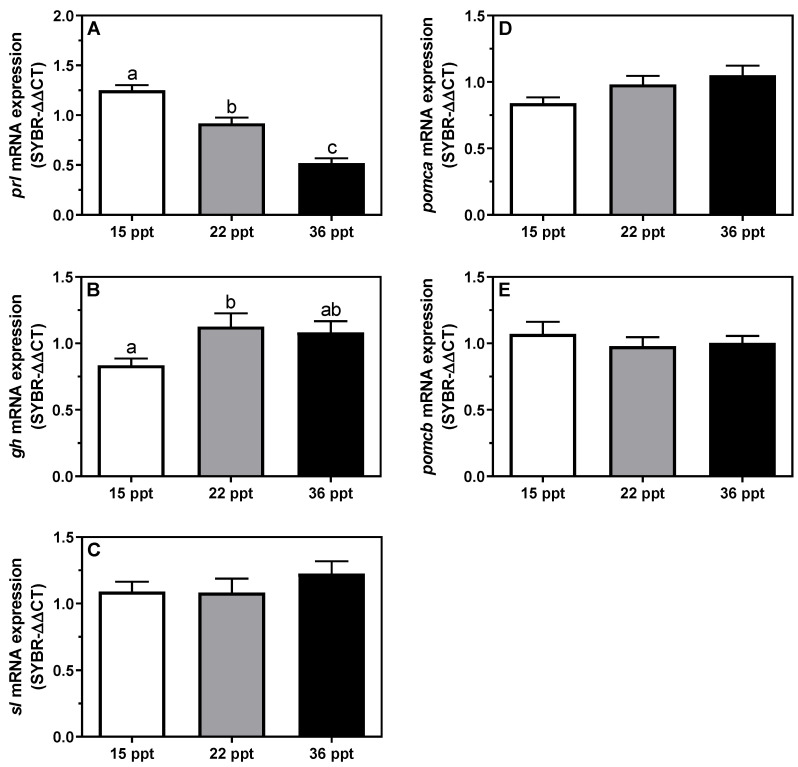
Changes in mRNA expression levels (relative to *actb* and *eef1a*) of adeno-hypophyseal (**A**) *prl*, (**B**) *gh*, (**C**) *sl*, (**D**) *pomca*, and (**E**) *pomcb* in *S. dumerili* individuals acclimated to different environmental salinities (15, 22, and 36 psu). Data are presented as mean ± SEM (n = 9–12). Different superscript letters indicate significant differences among salinities (*p* < 0.05, one-way ANOVA followed by Tukey´s *post hoc* test).

**Table 1 animals-11-02607-t001:** Specific primers used for real-time qPCR expression analyses, sizes of the amplified products, and GenBank accession numbers for respective genes.

Gene	Sequences (5′ → 3′)	Amplicon Size (bp)	GenBank Acc. No
*actb*	FW: CTCTTCCAGCCTTCCTTCCT	110	MW311085.1 XM_022757055.1
	RV: GTGTTGGCGTACAGGTCCTT		
*eef1a*	FW: GGAAGTTCGAGACCAGCAAG	144	MW311086.1 XM_022744048.1
	RV: CAGCCTCAAACTCACCAACA		
*gh*	FW: GACCCTGAACCAGAACCTGA	128	MW311089.1 XM_022769709.1
	RV: AGCGATGGAGAACAGATGCT		
*pomca*	FW: TGCATCCAGCTCTGTCACTC	136	MW311092.1 XM_022770028.1
	RV: TAGCCTGAGGTGAGGAGGA		
*pomcb*	FW: CTGGCCGGTCAGTTGGAGGG	107	MW311093.1 XM_022757428
	RV: ATACACACTGCCCTGTCTCT		
*prl*	FW: CAGAGGCAGACCTGTTGTCA	87	MW311094.1 XM_022747913.1
	RV: GCGTGTTAGCAGAGGAGGAC		
*sl*	FW: TATTTGCGTCGAGCTGTC	97	MW311095.1 XM_022746319.1
	RV: AAGAGGCAGCGAGGAATACA		

**Table 2 animals-11-02607-t002:** Plasma and hepatic parameters in *S. dumerili* individuals acclimated to different environmental salinities (15, 22, and 36 psu). Data are presented as mean ± SEM (n = 8–12). Values are expressed as grams of the wet weight of tissue (g^−1^w.w.). Different superscript letters indicate significant differences among salinities (*p* < 0.05, one-way ANOVA).

Parameters	15 Psu	22 Psu	36 Psu
Glucose (mM)	12.81 ± 0.89 ^a^	7.36 ± 0.77 ^b^	10.44 ± 0.39 ^a^
Lactate (mM)	5.20 ± 0.50 ^a^	4.81 ± 0.30 ^ab^	3.74 ± 0.23 ^b^
Triglycerides (mM)	2.01 ± 0.16	1.81 ± 0.20	1.87 ± 0.10
Glucose (mmol g^−1^ w.w.)	70.78 ± 4.58 ^a^	93.69 ± 4.52 ^ab^	103.30 ± 10.56 ^b^
Glycogen (mmol g^−1^ w.w.)	36.67 ± 4.75 ^a^	54.63 ± 4.18 ^ab^	61.98 ± 8.23 ^b^
Triglycerides (mmol g^−1^ w.w.)	48.96 ± 5.38 ^a^	67.49 ± 2.65 ^b^	51.05 ± 5.33 ^a^

## Data Availability

Data generated in this study are available on request from the corresponding author (Andre Barany; email: andre.barany@uca.es).

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
