# Peer review of "Osmoregulatory Plasticity of Juvenile Greater Amberjack (Seriola dumerili) to Environmental Salinity"

_animals, 2021, doi:10.3390/ani11092607_

Round 1

Reviewer 1 Report

In general the authors write a good osmoregulatory story for juvenile greater amberjack to environmental salinity. But some major issues to ask.

  1. Why the authors choose 15, 22, and 36 ppt as the general salinity, the part of Introduction is not clear, especially for 36ppt. We knew that the salinity of Mediterrean sea and Atlantic sea is different. Line 70-71 need more proofs.
  2. why the authors choose Na+/ K+- ATPase activity in the gill and the anterior and posterior regions of the intestine, not kidney (another osmoregulatory organ).
  3. How did you sample the gill and intestine? Gill include gill filaments and gill arches. How did you decide the intestinal (from anterior and posterior regions), please draw a figure to show, we know that intestine is a complex organ. Different part show different functions including anterior, middle, posterior intestine even rectum.
  4. For the qRT-PCR work, why just choose the only tissue-pituitaries ? Some hormone receptors in gill, intestine, kidney ... are not explored ?
  5. Cortisol as the stress indicator is not measured in this study.
  6. why the 22ppt is higher than other two conditions for muscle water content, the authors did not discuss well in the manuscript.
  7. why NKA activity is higher with increasing salinity in anterior intestine, the authors did not discuss well in the manuscript.
  8. Some significant genes should be worked in protein level using Western blot or immunohistology.

Reviewer 2 Report

Line 51. write belongs

Line 54. Rewrite. This species.....

Line 61. Its fast growth

Line 72. in RAS

Line 108. gills

Line 141. Are you sure that it was only 0.7 mL of 2-phenoxyethanol? This level of the narcotic was used by me in mugilids to just mildly anaesthetize the fry fish.

Line 352. delete "which"

Reviewer 3 Report

Review

Paper title: Osmoregulatory plasticity of juvenile greater amberjack (Seriola dumerili) to environmental salinity.

The authors studied the effects of different salinity regimes on physiological parameters and gene expression in juvenile greater amberjack. They found that the fishes can adapt to lower salinities without significant costs for their growth. These results provide important information on the biology of Seriola dumerili and may have important implication for its aquaculture.

All these reasons explain the relevance of the paper by Andre Barany and co-authors submitted to "Animals".

General scores.

The data presented by the authors are original and significant. All conclusions are justified and supported by the results. The study is correctly designed and technically sounds. In general, the statistical analyses are performed with excellent technical standards. We authors conducted careful work which will attract the attention of a wide range of specialists focused on aquaculture of greater amberjack.

Specific comments.

L 26. Change “ppt” to “psu” and throughout the text.

L 54. Change “This is species” to “It”

L 57. Change “seas” to “Seas”

L 67. Change “optimum” to “optimal”

L 76. Define “SW” here (L 82).

L 147. Change “metabolites” to “metabolite”

L 195. Change “spp” to “spp.”

L 263. Change “Previously,” to “Prior to analyzes”

L 449. Change “spp” to “spp.”

L 453. Change “spp.” to “spp.” twice

L 475. “Seriola dumerili” should be italicized.

Round 2

Reviewer 1 Report

-"Note that sometimes annotated genes encoding ion transport proteins in several species do not represent the original protein for which they are initially annotated. In most cases is because they are annotated just partially instead of the complete protein." So the relevence of hormone and hormone receptors must be discussed and how did you know sometimes annotated genes encoding ion transport proteins in several species do not represent the original protein for which they are initially annotated. In most cases is because they are annotated just partially instead of the complete protein. So some significant genes should be worked in protein level.

-The authors did not discuss well about the roles between anterior and posterior intestine. Some literatures demonstrated that the posterior intestine is the primary entrance of monovalent ions with high NKA activity, anterior intestine is a digestive site......The authors need to recalculate and discuss well from literatures.